# Kinetic Energy Calculation in Granite Particles Comminution Considering Movement Characteristics and Spatial Distribution

**Qing Guo [1,2], Yongtai Pan [1,2,*], Qiang Zhou [1,2], Chuan Zhang [1,2] and Yankun Bi [1,2]**

[1] School of Chemical and Environmental Engineering, China University of Mining and Technology (Beijing), Beijing 100083, China; tbp1600301001z@student.cumtb.edu.cn (Q.G.); bqt1700301034@student.cumtb.edu.cn (Q.Z.); bqt2000301006@student.cumtb.edu.cn (C.Z.); byanking945@gmail.com (Y.B.)

[2] Engineering Research Center for Mine and Municipal Solid Waste Recycling, Chemical Engineering and Technology, China University of Mining and Technology (Beijing), Beijing 100083, China

[*] Correspondence: panyongtai@cumtb.edu.cn; Tel.: +86-15010651331

**Abstract:** Profound knowledge of the movement characteristics and spatial distribution of the particles under compression during the crushing of rocks and ores is essential to further understanding kinetic energy release law. Various experimental methods such as high-speed camera technology, the coordinate method, and the color tracking method were adopted to improve the understanding of particles' movement characteristics and spatial distribution in rock comminution. The average horizontal velocities of the four size particles $\alpha$, $\beta$, $\gamma$, and $\delta$ are statistically calculated. The descending order of the particles' average velocity is $\gamma$, $\beta$, $\alpha$, and $\delta$. In comparison, the descending order of the particles' kinetic energy is $\alpha$, $\beta$, $\gamma$, and $\delta$. Moreover, the contribution of $\alpha$ particles to the total kinetic energy exceeds 70%. The spatial distribution characteristics of coarse and fine particles show different results. The probability of fine particles appearing in the range closer to the center area is greater, while the position of large particles appears to be more random. The color tracking results show that super-large particles generated by crushing are on the specimen's surface, while small particles are generally produced from inside. The above results indicate a connection between the particle generation mechanism, movement characteristics, and spatial distribution in the comminution process.

**Keywords:** brittle materials; uniaxial compression; comminution; particle size; movement characteristics; particle velocity; kinetic energy; spatial distribution

## 1. Introduction

The problem of dynamic fragmentation is a scientific field that has been unresolved for a long time. Compared with the quasi-static fracture of plastic materials, a dynamic fracture is more difficult to understand [1–3]. Dynamic fracture is challenging to study because this process involves complex interactions over an extensive period and space. The main hazard of dynamic fracture is the kinetic energy carried by the ejected fragments during the occurrence. The speed of the destruction of the block sometimes even exceeds 1000 m/s, which is extremely harmful to human activities and the natural environment. [4–6]. The compression and fragmentation of brittle materials are not limited to impact loading. Under the action of the quasi-static compression load, ceramic specimens can still undergo "explosive" damage [7]. Since the research by Mott [8], the dynamic fracture and fragmentation of solids have been a hot research topic. The dynamic fracture of brittle materials can be studied by the uniaxial compression test [9,10], conventional triaxial unloading test [11], true triaxial rock-burst test [12,13], and high-speed impact test [14,15].

Among them, the traditional uniaxial compression and triaxial tests have lower loading rates, which are generally considered to be quasi-static loading, while split Hopkinson pressure bar (SHPB) loading and high-speed impact tests are dynamic loadings [10,16]. Except for conventional triaxial tests restricted by hydraulic cylinders, dynamic fragmentation can be observed in other loading conditions. The most commonly used observation instrument is a high-speed camera that can track particle trajectories and speed measurement [17].

The particle tracking dynamic system can realize the movement tracking of complex and large numbers of particles. This technology is mainly used in high-speed impact tests [18]. The laboratory conducts dynamic fracture experiments of brittle materials to study phenomena such as rock bursts, volcanic eruptions, earthquakes, and planetary collisions. Commonly used experimental materials are basalt [19], quartz [20,21], sandstone, etc. [18]. The research focuses on the particle velocity distribution after dynamic fracture [15], fragment size [10], rebound angle [14], etc.

Energy evolution is a common method for studying dynamic fracture. The quasi-static loading method calculates the input energy through the load-displacement curve [22], and the SHPB loading calculates the absorbed energy of the specimen through the incident and transmitted waves [23]. The high-speed dynamic experiment considers that the kinetic energy of the bullet is input energy [14].

The speed of broken particles can be measured by image tracking technology, and the kinetic energy can be calculated by weighing the particles. Based on the law of conservation of energy, the dissipative heat energy generated by the force-heat coupling process can be studied [24]. Xie [22,25,26] found that studying the energy dissipation and energy release of rock mass structures from the perspective of macroscopic energy conservation can be used to estimate the splash velocity of fragmented rock blocks. Li et al. [10] used SHPB to study the dynamic crushing particle size characteristics, fragment distribution and crushing laws of rock materials. Rait et al. [27] used the discrete element method to study the effect of the loading rate on static fracture and dynamic fracture and analyzed the relationship between the kinetic energy and frictional energy dissipation during the comminution process. Wang et al. [28] studied the energy distribution during the quasi-static confined comminution of granular materials. Xiao et al. [29] analyzed and compared the energy dissipation law of carbonate sand quasi-static and dynamic compression. Zhang [30] studied the average fragmentation and velocity of the debris under a quasi-static load of brittle materials, which agree with the theoretical calculations. The above research mainly focused on the average particle size and velocity and did not involve the velocity and kinetic energy distribution of the characteristic particle size. Exploring the dynamic fracture mechanism of brittle materials requires in-depth research on the speed, kinetic energy, and temporal and spatial distribution characteristics of particles of different sizes produced by crushing.

In response to the above problems, this paper uses high-speed camera technology and digital image motion analysis software to study the velocity–size relationship of particles produced by uniaxial compression crushing of granite and the contribution of products of different sizes to kinetic energy. The coordinate method is used to study the spatial characteristics of fragment distribution at different scales. The color tracking method is used to study the relationship between the spatial characteristics of the fragment distribution and the generation location. The research methods and results have positive significance for describing the splash particles' temporal and spatial characteristics and revealing the kinetic energy release law of the dynamic fracture of brittle materials. At the same time, it is of positive significance for the quantitative calculation of dissipative heat energy and the study of energy evolution in the comminution process.

## 2. Materials and Methods

### 2.1. Experimental Materials

The granite was selected from Queshan County, Zhumadian, and all samples were cut and processed from a relatively complete ore body. Firstly, a cylindrical core with a diameter of 50 mm was drilled, and then a cylindrical specimen with a height of 100 mm was cut. A total of 15 granite specimens were prepared in this experiment, as shown in Figure 1. The stone grinder and sandpapers were used to grind both ends of the test piece carefully so that the parallelism of the upper and lower surfaces was within 0.05 mm, and the surface flatness was within 0.02 mm. The samples had good integrity and uniformity, and the average uniaxial compressive strength was 110 MPa. The X-ray fluorescence (XRF) test shows that $SiO_2$ has the highest content in granite, and the detailed content of other substances is shown in Table 1.

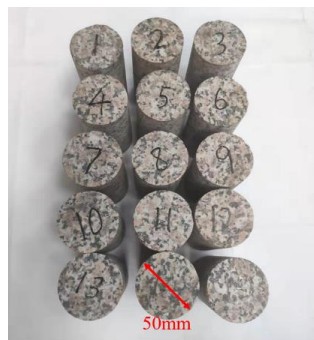

**Figure 1.** Granite specimens.

**Table 1.** Granite mineral content.

| $SiO_2$% | $Al_2O_3$% | $Na_2O$% | $K_2O$% | CaO% | $Fe_2O_3$% | MgO% | $TiO_2$% |
|---|---|---|---|---|---|---|---|
| 67.75 | 15.66 | 4.81 | 3.84 | 2.73 | 2.49 | 1.41 | 0.318 |

### 2.2. Experimental System

The uniaxial compression test of granite specimens was carried out using the method of force loading. The experimental loading rates were 1, 2, 3, 4, 5 kN/s, with five loading rates and three tests for each loading rate. The unloading process had the same rates as the loading process. This test uses the TAW-3000 hydraulic servo test system (Changchun City Chaoyang Test Instrument CO., LTD., Changchun, China) (as shown in Figure 2a). The testing machine has a portal frame with a stiffness greater than 5 GN/m, which can provide an axial force of 3000 kN and a resolution of 20 N. The resolution of the axial deformation of the specimen is 0.5 μm. The high-speed camera used in this experiment has a shooting frequency of 800 Hz and a shooting area of 400 mm × 500 mm, which is used to record the horizontal velocity of the broken particles' movement. The focal length of the lens used in this experiment was 50 mm.

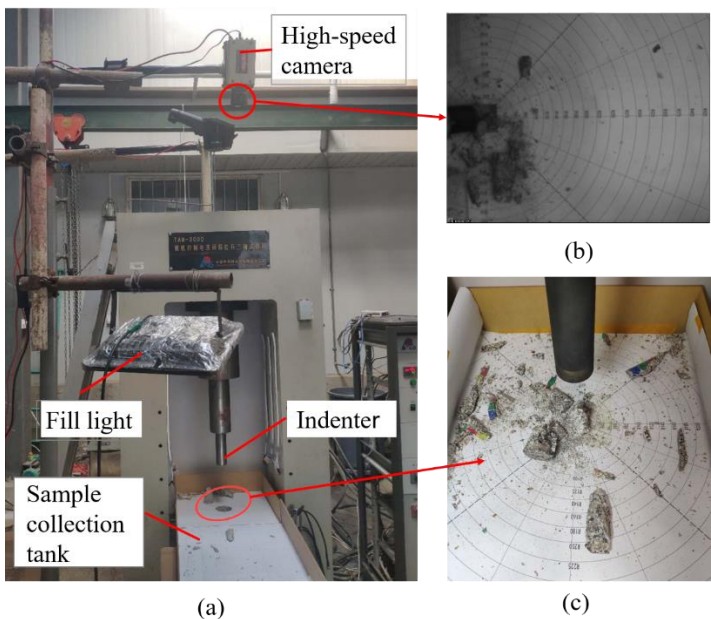

**Figure 2.** Uniaxial compression crushing dynamic capture system (**a**) Uniaxial compression loading system (**b**) High-speed camera images (**c**) Spatial distribution of fragments.

## 3. Results and Discussion

### 3.1. Force-Displacement Relationship of Uniaxial Compression

In the process of the uniaxial compression of the specimen, the displacement of the indenter changed with the load. This change process is usually divided into four stages [31]: the crack compaction stage, elastic stage, microcrack stable-growth stage, and the unstable cracking stage. The accelerated expansion phase and the post-peak segment are shown in Figure 3a. At the same time, energy accumulates, dissipates, and releases inside the specimen. Regardless of the heat exchange between the specimen and the environment, the relationship between input energy, elastic energy, and dissipation energy is as follows [32]:

$$U = U^d + U^e \tag{1}$$

where $U$ is the work done by the external force on the rock, i.e., the energy absorbed by the rock; $U^d$ is the energy dissipated by the rock during the loading process, which is mainly used for the internal damage and plastic deformation of the rock; and $U^e$ is the elastic strain energy stored in the rock. The value of elastic energy can be determined by the area of the unloading curve and the coordinate axis, as shown in Figure 3b. According to the above calculation method, the input energy of specimen 11 before failure is 47.16 J, of which the elastic energy accounts for 24.43 J, with a compression displacement of 0.302 mm.

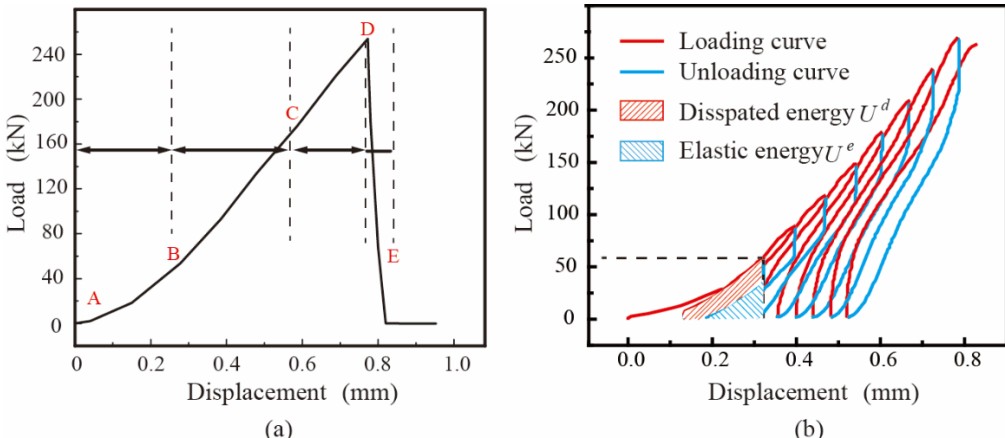

**Figure 3.** The uniaxial compression load-displacement curve (**a**) Different stages of uniaxial compression (**b**) Calculation of elastic energy and dissipation energy.

### 3.2. Characteristics of Uniaxial Compression Failure Fragments

In order to facilitate the analysis and study of the movement characteristics of different sizes of uniaxial destruction fragments (at the same time limited by the camera resolution), the fragments obtained after the uniaxial compression experiment were divided into four groups according to the particle size, namely $\alpha$ particles, $\beta$ particles, $\gamma$ particles and $\delta$ particles [33]. The size of the fragments were divided into +13 mm, 6–13 mm, 3–6 mm, and −3 mm, as shown in Figure 4. Since the fragments were often irregular, the sieving result was used as the measurement and calculation standard during measurement.

The following information can be obtained through observation and analysis of high-speed photography images (Figure 5). In the early stage of macro-destruction, the smaller particles ($\gamma$ particle) were ejected from the surface of the specimen first. Such particles are located at the front of the detrital cluster and move extremely fast. In the early stage of macro-destruction, the largest particles ($\alpha$ particle) peeled off the surface of the specimen. These particles are located in the front and middle part of the detrital cluster and move faster. In the middle stage of the macro destruction, the larger particles ($\beta$ particle) peeled off from the surface of the specimen. Such particles are located in the middle of the detrital cluster and move slowly. At the end of macro destruction, the smallest particles ($\delta$ particle) were produced, which are located at the back of the detrital cluster and move very slowly. The generation time, spatial location and movement characteristics of the four types of particles were summarized in Table 2.

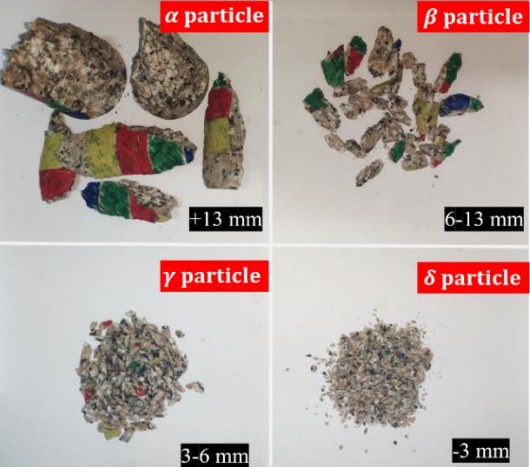

**Figure 4.** Classification of fragments produced by uniaxial compression.

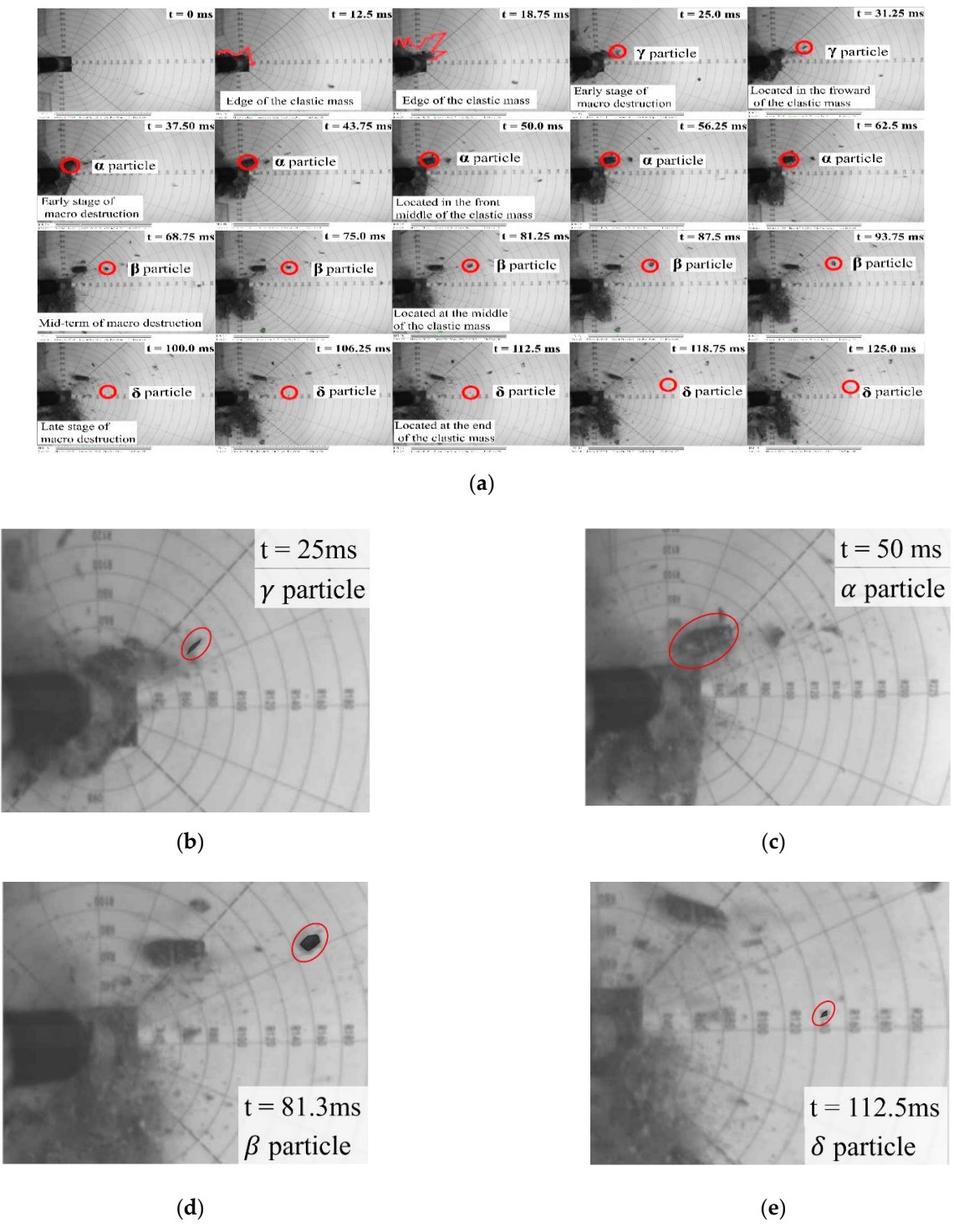

**Figure 5.** The temporal and spatial characteristics of the movement of fragments particles. (**a**) High-speed camera images in time series (**b–e**) The enlarged view of the four types of particles.

**Table 2.** Temporal and spatial position and movement characteristics of fragments particles.

| Particle Type | Size/mm | Generation Time | Spatial Location | Movement Characteristics |
|---|---|---|---|---|
| α | +13 | Early stage of macro destruction | Front middle of the detrital cluster | Surface peeling, ejection, Roll along the length |
| β | 6–13 | Early and mid-term macro destruction | Middle of detrital cluster | Surface peeling, rotating |
| γ | 3–6 | Early stage of macro destruction | Forward of the detrital cluster [34] | Ejection, extremely fast |
| δ | −3 | Mid- to late period of macro destruction | The tail of the detrital cluster | Friction occurs, slower |

### 3.3. Fragments Velocity Characteristics

According to the classification characteristics of Section 3.2, the tracking function of high-speed photography is used to count the horizontal velocity of each sample produced by the representative α, β, γ, and δ particles. In each specimen, about 10 particles were selected as representatives for each of the four particle types. (the super-large particles may be less than 10). The particle size in high-speed photography is measured by the calibration function in the video viewing software. The velocity of the four types of particles in specimen 11 is shown in Table 3. As shown in Table 4, in terms of the average velocity, the descending order is $v_{A\gamma}$, $v_{A\beta}$, $v_{A\alpha}$, and $v_{A\delta}$.

**Table 3.** Four types of particle velocity of specimen 11.

| Serial Number | The Velocity of Particles m/s | | | |
|:---:|:---:|:---:|:---:|:---:|
| | α Particle | β Particle | γ Particle | δ Particle |
| 1 | 14.75 | 8.11 | 13.28 | 2.23 |
| 2 | 4.86 | 8.25 | 15.08 | 1.94 |
| 3 | 8.58 | 6.55 | 14.97 | 2.39 |
| 4 | 4.33 | 8.44 | 12.15 | 2.42 |
| 5 | 3.78 | 8.59 | 13.09 | 1.64 |
| 6 | 6.77 | 6.58 | 7.63 | 1.74 |
| 7 | 6.13 | 7.59 | 7.71 | 1.83 |
| 8 | 6.11 | 7.91 | 6.63 | 1.92 |
| 9 | 7.27 | 6.83 | 6.40 | 2.56 |
| 10 | 6.42 | 7.09 | 7.90 | 1.70 |
| $v_A$ | 6.900 | 7.594 | 10.483 | 2.036 |
| STD. | 2.943 | 0.740 | 3.358 | 0.317 |

Note: STD. is the abbreviation of standard error values.

**Table 4.** Four types of particle velocities in different specimens.

| Specimen Number | The Average Velocity of Particles m/s | | | |
|:---:|:---:|:---:|:---:|:---:|
| | α Particle | β Particle | γ Particle | δ Particle |
| 1 | 3.052 | 5.827 | 9.913 | 1.403 |
| 2 | 4.945 | 5.151 | 8.009 | 2.826 |
| 3 | 2.480 | 2.825 | 4.972 | 1.278 |
| 4 | 2.111 | 2.104 | 8.797 | 1.476 |
| 5 | 4.194 | 4.052 | 7.821 | 3.583 |
| 6 | 2.950 | 4.864 | 7.263 | 2.074 |
| 7 | 4.238 | 4.856 | 8.206 | 2.634 |
| 8 | 2.015 | 3.604 | 5.530 | 1.313 |
| 9 | 3.154 | 3.160 | 6.511 | 1.718 |
| 10 | 6.017 | 6.357 | 18.094 | 2.830 |
| 11 | 6.900 | 7.594 | 10.483 | 2.036 |
| 12 | 5.163 | 6.834 | 9.829 | 2.162 |
| 13 | 6.519 | 5.434 | 10.068 | 2.489 |
| 14 | 2.404 | 4.109 | 7.228 | 1.050 |
| 15 | 2.254 | 4.909 | 8.749 | 1.822 |
| $v_A$ | 3.893 | 4.779 | 8.765 | 2.046 |
| STD. | 1.615 | 1.467 | 2.949 | 0.691 |

### 3.4. Mass Distribution of Fragments

The average value of the horizontal velocity of the four types of particles in 15 groups of specimens is taken as the velocity benchmark for calculating the kinetic energy. The

key to calculating kinetic energy is to establish the corresponding relationship between speed and mass. Due to the limited field of view of high-speed photography, it is impossible to match the particles flying on the screen with the particles still in the tray. Therefore, it can only be analyzed by collecting the speed of particles of different characteristic sizes flying through the field of view to form statistical data. The mass of particles with characteristic sizes can be obtained by sieving. Figure 6 shows the sieving data of the four areas—I, II, III, and IV—of specimen 11. The positions of the four zones are shown in Figure 7.

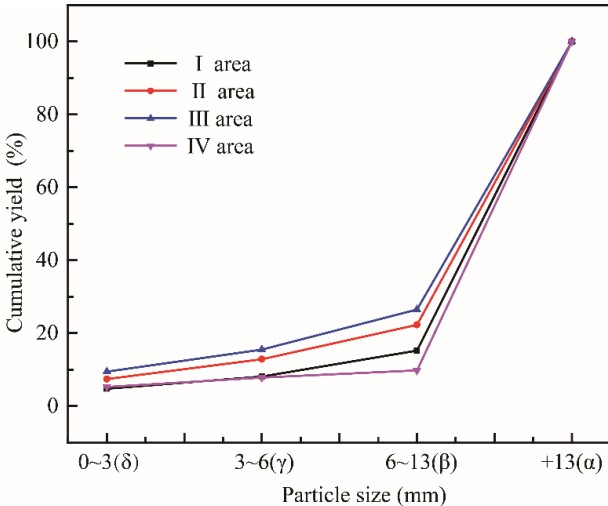

**Figure 6.** Particle size distribution curve of fragments in different areas of specimen 11.

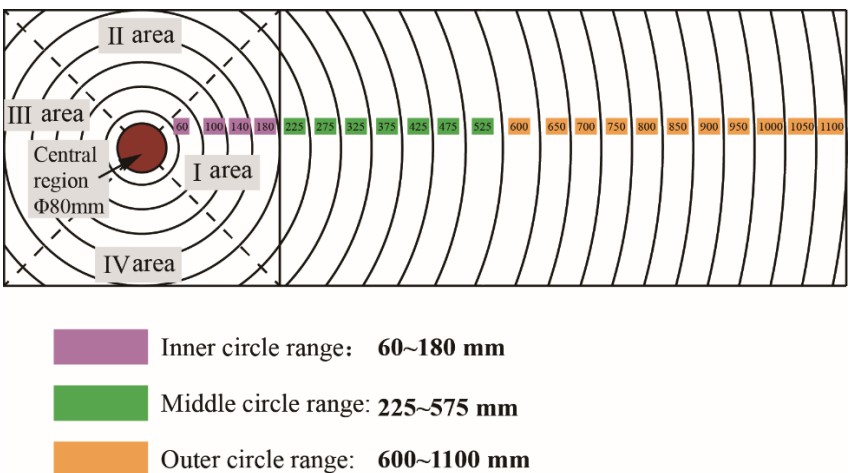

**Figure 7.** Division of the spatial distribution of fragments.

Theoretically, the distribution of fragments in the four areas after the uniaxial compression failure of homogeneous materials is the same. However, due to the differences in the internal cracks of the materials, the mass distribution of different specimens after crushing is random. Figure 8 shows the proportion of fragments in each area after crushing the five groups of specimens. In most cases, the central area accounts for the largest proportion, with an average mass proportion of 45%. The loading rate variation range of the center area mass between 1–4 kN shows a decreasing trend with the loading rate increase. The mass proportions of the remaining four regions show strong randomness in a single experiment, with an average mass proportion of 10 to 20%. If the four peripheral areas are regarded as a whole, it is opposite to the changing trend of the mass of the central area, and its total mass shows a law of increasing with the increase of loading rate.

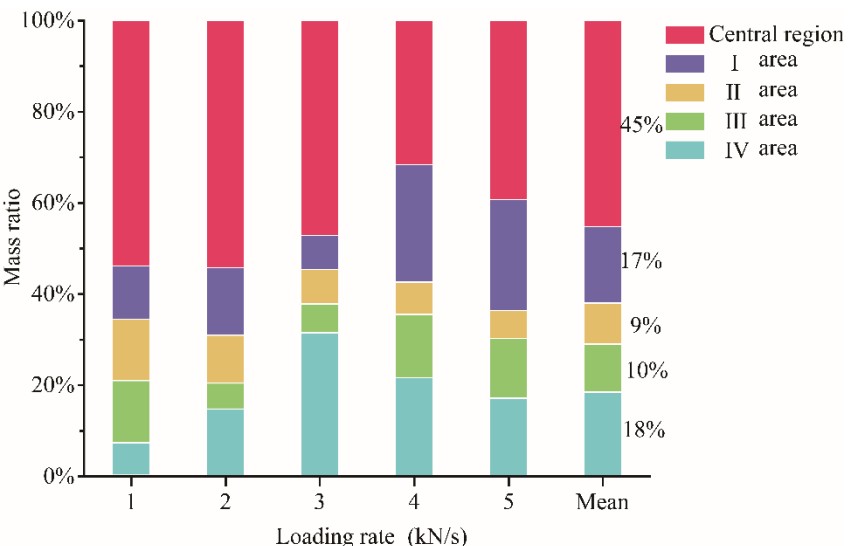

**Figure 8.** Mass distribution of fragments partition.

*3.5. Kinetic Energy of Single-Axis Destruction Fragments*

According to the average velocity of the four types of particles and the size distribution of the fragments in each area, the total kinetic energy of each specimen was calculated by using Equation (2):

$$E_k = \frac{1}{2}\left(\sum m_\alpha v_{A\alpha}^2 + \sum m_\beta v_{A\beta}^2 + \sum m_\gamma v_{A\gamma}^2 + \sum m_\delta v_{A\delta}^2\right) \qquad (2)$$

where $\sum m_\alpha$ is the sum mass of $\alpha$ particles, $v_{A\alpha}$ is the average velocity values of the $\alpha$ particles. Correspondingly, other symbols indicate the total mass and average velocity of various particles of $\beta$, $\gamma$, $\delta$.

Table 5 shows the kinetic energy released by each area in the crushing process and the proportion of kinetic energy corresponding to various particles. Sum the total kinetic energy of each area to obtain the total kinetic energy of 5.184 J released by the crushing of specimen 11. Similarly, the kinetic energy released by the crushing of other specimens can be calculated. According to the calculation method described in 3.1, the input energy and elastic energy data of each test piece are calculated, as shown in Table 6. For specimen 11, the ratio of input energy into kinetic energy is 10.79%, and the ratio of elastic energy into kinetic energy is 20.84%. The average of the ratio of kinetic energy to elastic energy of all specimens is 16.03%, and the average of the ratio of kinetic energy to input energy is 7.92%.

As there are few studies on the kinetic energy calculation of the rock fragmentation under uniaxial compression, the author has not found convincing data to verify it. However, in similar destruction modes, the proportion of kinetic energy can be used as evidence. For example, the impact of spherical particles [35], rock blasting [36] and the ratio of kinetic energy to input energy are approximately 3% and 3–21%, respectively. In the true triaxial failure of brittle rocks [37], the ratio of kinetic energy to elastic energy ranges from 8 to 50%. In dynamic fracture of pre-cracked rock specimens, the SHPB system was used, and the ratio of kinetic energy to input energy ranges from 22 to 59% [38].

The kinetic energy of the four kinds of particles generated by the crushing of the specimen at different loading rates is shown in Figure 9. It can be seen that the total kinetic energy increases with the increase of the loading rate within the range of loading rate of 1–4 kN/s. Furthermore, the main factor affecting the total kinetic energy is the kinetic energy of $\alpha$ particles. The relationship between the kinetic energy of other types of particles and the loading rate is not obvious. When the loading rate is 5 kN/s, the total kinetic energy decreases, which is mainly affected by the decrease of the kinetic energy of $\alpha$ particles.

**Table 5.** Fragments kinetic energy in different areas of specimen 11.

| Particle Type | I Area | | II Area | | III Area | | IV Area | |
|---|---|---|---|---|---|---|---|---|
| | $E_k$/mJ | PCT.% | $E_k$/mJ | PCT.% | $E_k$/mJ | PCT.% | $E_k$/mJ | PCT.% |
| $\alpha$ | 927.51 | 73.99 | 993.57 | 64.76 | 818.92 | 62.35 | 945.77 | 87.28 |
| $\beta$ | 167.85 | 13.39 | 249.76 | 16.28 | 235.87 | 17.96 | 38.06 | 3.51 |
| $\gamma$ | 150.28 | 11.99 | 276.82 | 18.04 | 244.18 | 18.59 | 92.59 | 8.54 |
| $\delta$ | 7.94 | 0.63 | 14.13 | 0.92 | 14.49 | 1.10 | 7.15 | 0.66 |
| Total $E_k$ | 1253.58 | 100 | 1534.29 | 100 | 1313.45 | 100 | 1083.57 | 100 |

Note: PCT. is the abbreviation of percent.

**Table 6.** Input energy, elastic energy and kinetic energy of different specimens.

| Specimen Number | $U$  J | $U^e$  J | $E_k$  mJ | $E_k/U^e$  % | $E_k/U$  % |
|---|---|---|---|---|---|
| 1 | 56.21 | 30.87 | 1435.36 | 5.21 | 2.55 |
| 2 | 40.53 | 23.09 | 2414.18 | 13.36 | 5.96 |
| 3 | 25.39 | 12.53 | 626.53 | 5.85 | 2.47 |
| 4 | 33.78 | 15.88 | 1073.90 | 6.21 | 3.18 |
| 5 | 27.65 | 12.36 | 2281.99 | 18.50 | 8.25 |
| 6 | 44.28 | 25.69 | 1441.39 | 6.11 | 3.25 |
| 7 | 23.72 | 8.38 | 3481.25 | 30.44 | 14.68 |
| 8 | 25.98 | 14.05 | 823.46 | 5.98 | 3.17 |
| 9 | 26.09 | 10.62 | 1793.53 | 16.06 | 6.88 |
| 10 | 41.20 | 20.98 | 4369.55 | 25.84 | 10.61 |
| 11 | 47.16 | 24.43 | 5184.90 | 21.22 | 10.99 |
| 12 | 21.01 | 8.56 | 2542.20 | 30.55 | 12.10 |
| 13 | 27.90 | 11.87 | 7289.66 | 54.95 | 26.13 |
| 14 | 30.01 | 13.75 | 1557.10 | 9.01 | 5.19 |
| 15 | 41.37 | 20.81 | 1383.43 | 7.87 | 3.34 |
| Average | 34.15 | 16.25 | 2604.32 | 16.03% | 7.92% |

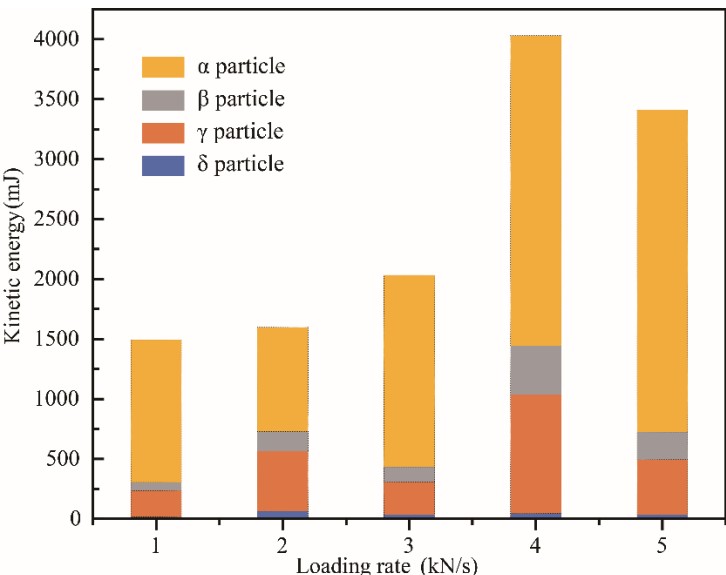

**Figure 9.** The kinetic energy of four particles under different loading rates.

The proportion of kinetic energy of various particles in specimen 11 can be obtained by summarizing the data in Table 5. The distribution law of kinetic energy can be seen in Figure 10; the kinetic energy proportions of the four types of particles are ranked from high to low as $E_{K\alpha} > E_{K\gamma} > E_{K\beta} > E_{K\delta}$. The kinetic energy of $\alpha$ particles accounts for about 70%, the kinetic energy of $\gamma$ particles accounts for close to 20%, the kinetic energy

of β particles accounts for close to 10%, and the kinetic energy of δ particles accounts for about 1.5%. Comparing the average values of specimen 11 and specimens 1 to 15 shows that the kinetic energy distribution of various particles of a single specimen is not significantly different from the overall distribution. The two indicators that affect the magnitude of kinetic energy are speed and quality. The $\alpha$-type particles have the largest mass, and the $\gamma$ particles have the largest velocity. Since the mass of alpha particles is more than an order of magnitude higher than that of gamma particles, the speed of $\gamma$ particles is several times that of alpha particles. This has led to massive particles becoming the main contributor to kinetic energy.

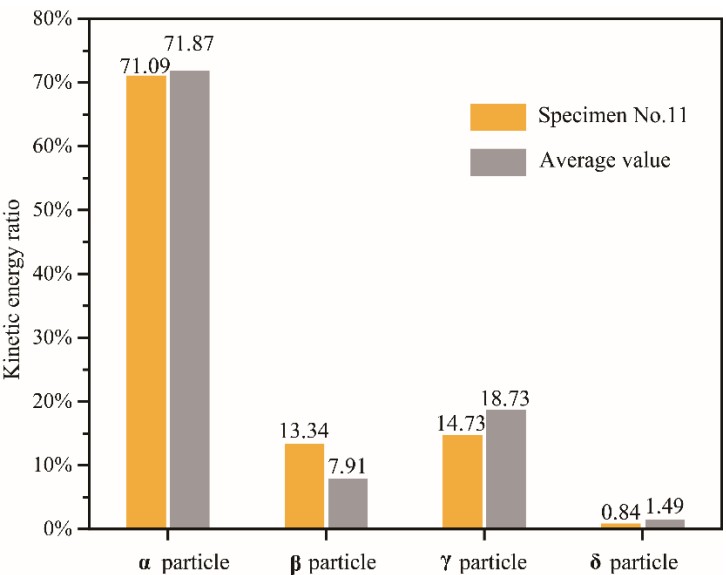

**Figure 10.** Percentage of the kinetic energy of various particles.

### 3.6. Spatial Distribution of Fragments

With a diameter of 6 mm as the standard, the fragments were divided into large particles and fine particles. The masses of the fragments in the I area and the extended area along the radial direction were counted after crushing. Figure 11a shows the spatial distribution of fine particles generated by the crushing of as there are few studies on the all specimens. The density of the data points represents the possibility of corresponding mass fine particles in the corresponding area. The blank area near the center area indicates that the mass of fine particles produced in the inner circle is more considerable, generally above 0.1 g. The closer to the center area, the more fine particles. Figure 11b shows the changing trend of the total mass of fine particles of specimens 1 to 15 in the range of 60 to 200 mm, which confirms this rule. Figure 11b shows that the particle mass has a maximum value at 300 mm. This aggregation phenomenon reflects the fine particle velocity distribution characteristics, which represents the intersection of the $\gamma$ and δ particles. After the maximum point, the mass of fine particles decreases as the distance increases, and the decreasing trend gradually slows down. In the area larger than 1000 mm, the particle mass tends to increase again, mainly because the collection trough restricts fragment movement. The loading rate has no significant effect on the spatial distribution of fine particles.

The spatial distribution of large particles shows strong randomness, and the probability of super large particles is small. It can be seen from Figure 12a that there are only four particles larger than 70 g in all of the data, but they contribute most of the mass of the inner circle and the middle circle. Among more than 300 sets of data, there were only 16 sets of super-large particles with a mass greater than 20 g. There were only four groups of super large particles in the outer circle, and the mass was less than 50 g. From the spatial distribution of large particles, it can be seen that the input energy of this uniaxial compression and crushing is limited, which is not enough to push the super large particles to

a more distant area. The input energy may be related to the material properties and loading rate, and it is worthy of further exploration. For 15 sets of experiments, larger loading rates are more likely to produce large splashing particles. As shown in Figure 12b, the mass of large particles produced by specimens 10–15 (loading rate 4–5 kN/s) is larger than that of specimens 1–9 (loading rate 1–3 kN/s).

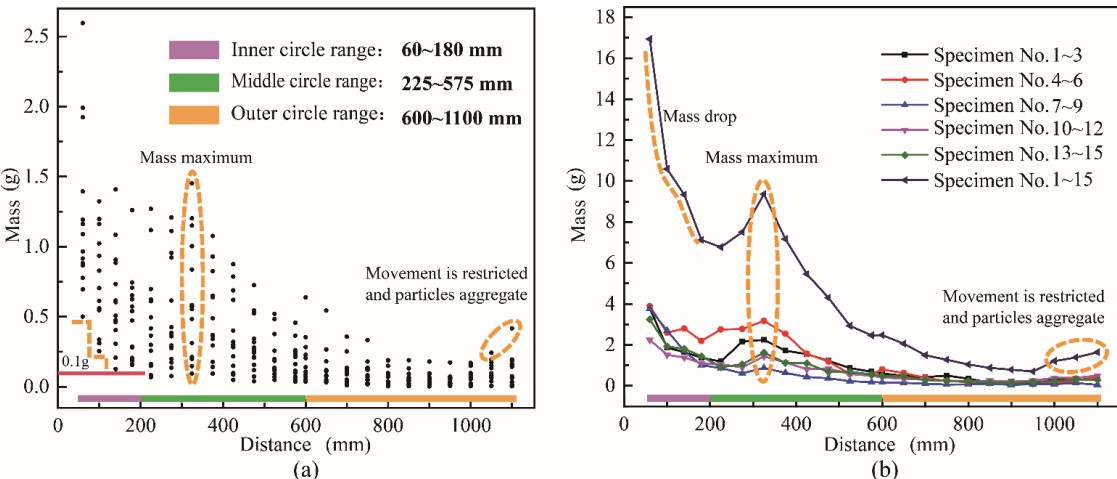

**Figure 11.** Spatial distribution characteristics of fine particles. (**a**) Scatter plot of particle spatial distribution (**b**) Summary of particle spatial distribution of each group.

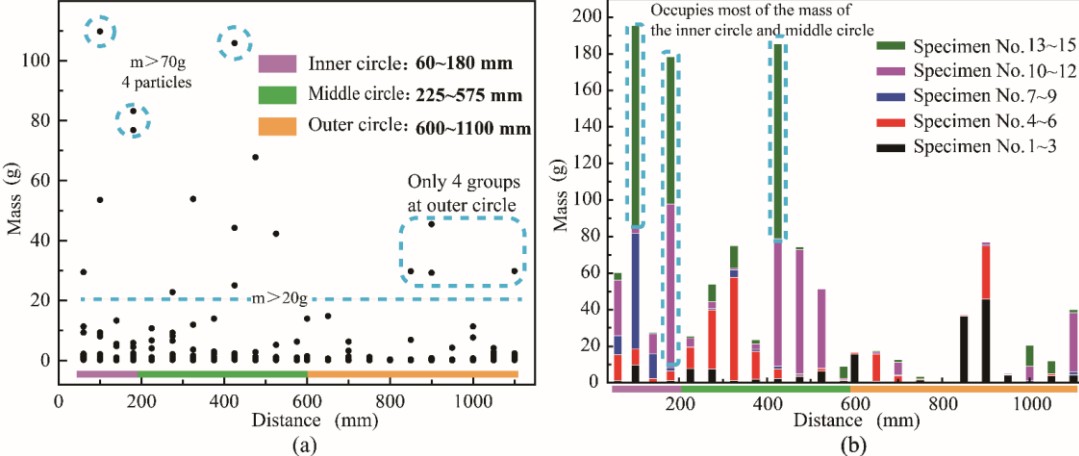

**Figure 12.** Spatial distribution characteristics of large particles. (**a**) Scatter plot of particle spatial distribution (**b**) Summary of particle spatial distribution of each group.

### 3.7. Location of Fragments

The surface of the specimen was painted. The debris larger than 6 mm can be divided into surface particles and internal particles according to whether there is a color on the surface. The mass of the two types of particles in area I along the radial direction is counted. On the whole, there is no apparent difference between Figures 12 and 13, which indicates that most of the particles larger than 6mm are surface particles; that is, at least one surface is the surface of the test specimen. On the contrary, it is easier to compare the difference between Figures 12 and 13 from the spatial mass distribution of the internal particles. That is to say, the mass of the particles at each distance in Figure 14b is the difference between the mass of the particles at the corresponding distance in Figures 12 and 13. Therefore, although Figures 12b and 13b are relatively close in morphology, there are differences in the number and mass of particles.

This part of the difference represents the particles generated from the inside of the specimen. From the mass distribution, it can be found that the mass of the internal particles in this part is small, and the distribution characteristics are similar to the mass-spatial distribution of fine particles in Figure 11. The mass spatial distribution of internal particles presents the following law as a whole: the mass near the center is large, the mass decreases rapidly as the distance increases, and the rate of decrease gradually decreases. Simultaneously, such particles' appearance will still show a certain degree of randomness, and there may even be no particles in some areas. All in all, the spatial distribution characteristics of particles larger than 6 mm generated inside have a part of the characteristics corresponding to particles smaller than 3 mm and particles larger than 6 mm on the surface of the specimen, which belong to the transition type between the two.

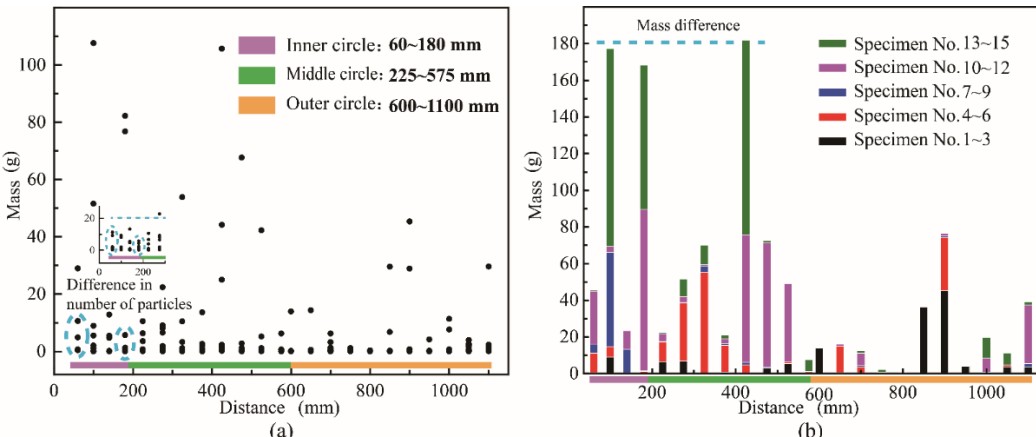

**Figure 13.** Spatial distribution characteristics of surface particles. (**a**) Scatter plot of particle spatial distribution (**b**) Summary of particle spatial distribution of each group

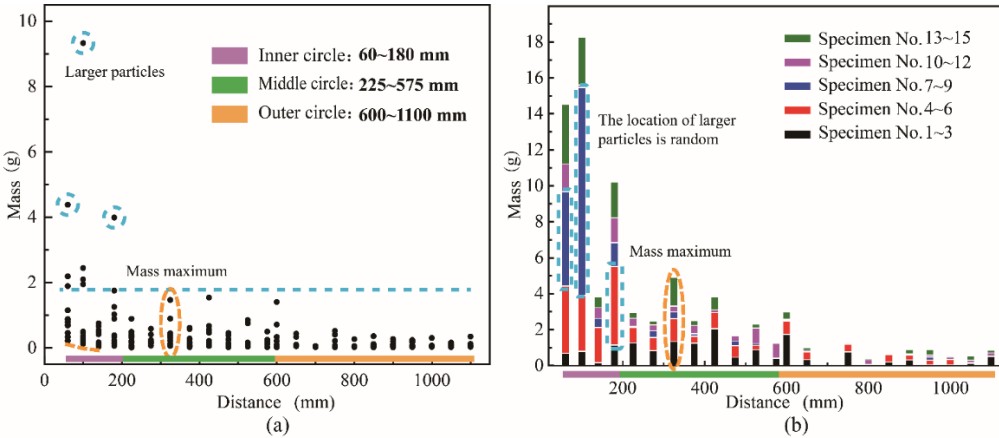

**Figure 14.** Spatial distribution characteristics of internal particles. (**a**) Scatter plot of particle spatial distribution (**b**) Summary of particle spatial distribution of each group

## 4. Conclusions

The fragments are divided into four types of particles according to the particle size.

The average horizontal velocities of the four size particles α, β, γ, and δ are statistically calculated. The descending order of the particles' average velocity is γ, β, α, and δ. Since the mass difference of different types of particles is greater than the influence of the velocity difference on kinetic energy, the descending order of the particles' kinetic energy is α, β, γ, and δ. Among them, the contribution of alpha particles to the total kinetic energy exceeds 70%. The loading rate has little effect on the particle velocity. When the loading rate is higher, more alpha particles leave the central area, resulting in more input energy

being converted into kinetic energy. The percentage of input energy converted into kinetic energy of specimen 11 is 5.9% during the crushing process.

The spatial distribution characteristics of large particles and fine particles were analyzed by the coordinate method. As a result, it was found that there was a greater probability of fine particles appearing in the range closer to the central area; this reflects that most of the fine particles have a lower velocity. The maximum value of the fine particles' mass appears in the middle circle, which indicates that there are also particles with higher speed in the fine particles, namely $\gamma$ particles. These kinds of particles overlap with the slower particles, causing the phenomenon of mass maximum. The locations of large particles are random, but they are more likely to appear within the middle circle. A larger loading rate can produce more large splashing particles, which is consistent with the kinetic energy characteristics of the loading rate.

The color tracking method was used to study the location of particles larger than 6 mm in the specimen. It was found that at least one surface of the super large particles produced by crushing was the surface of the test specimen. Those particles produced entirely from the inside of the specimen are relatively small and have similar spatial distribution characteristics to fine particles. Therefore, it can be judged that fine particles and particles of smaller size are generally generated by friction between the cross-sections of the specimen when the specimen is broken. The speed of such particles is generally low. Most of the large particles and a few small particles are directly peeled off the surface of the broken specimen and have a higher splash speed.

**Author Contributions:** Y.P. provides overall experimental ideas and methods; Q.G. collected and archived the experiment results; Q.G. analyzed the experimental data and initiated the writing of the paper; Q.Z., C.Z. and Y.B. help analyze the experimental data. All authors have read and agreed to the published version of the manuscript.

**Funding:** The financial support from National Natural Science Foundation of China (Grant No. 52074308).

**Conflicts of Interest:** The authors declare no conflict of interest.

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
