# Peer review of "Kinetic Energy Calculation in Granite Particles Comminution Considering Movement Characteristics and Spatial Distribution"

_minerals, doi:10.3390/min11020217_

Round 1

Reviewer 1 Report

This manuscript studies kinetic energy calculation in granite particles comminution. The authors used high-speed camera technology and digital image motion analysis software to study the relationship of particles size and kinetic, and the relationship between the spatial characteristics of the fragment distribution and the generation location. This is an interesting topic and the manuscript could potentially be a good paper. However, some clarifications on the methodology are expected.

The dimension of specimen is not clear.  Author mentioned “a cylindrical sample with a height of 100 mm is cut into a total of 15 granite samples” and “Use a stone grinder and sandpaper 94

to grind both ends of the test piece carefully so that the parallelism of the upper and lower surfaces is within 0.05 mm”. I am confused about the exact size of the specimen.

Load-displacement curve is missing. Is Figure 3 the load-displacement curve for specimen No.11? Why do the Figure 3(a) and (b) show different behavior, i.e. ductile vs brittle?

The author mentioned “According to the above calculation method, the input energy of specimen No.11 before failure is 47.162 J, of which the elastic energy accounts for 24.430 J.” However, the corresponding displacement is not is not given.

The English of this manuscript should be improved.

How do you consider the energy related to rotational velocity?

The camera can only capture the velocity in 2 directions, while missing the velocity in the direction perpendicular to the camera. How do you consider this issue? You should clearly state the limitations of the method.

The energy distribution in quasi-static comminution should be also reviewed, such as [1] Wang, P., Arson, C. Energy distribution during the quasi-static confined comminution of granular materials. Acta Geotech. 13, 1075–1083 (2018). https://doi.org/10.1007/s11440-017-0622-5 [2] Xiao, Y., Yuan, Z., Chu, J. et al. Particle breakage and energy dissipation of carbonate sands under quasi-static and dynamic compression. Acta Geotech. 14, 1741–1755 (2019). https://doi.org/10.1007/s11440-019-00790-1

Author Response

The authors thank the reviewers for the most helpful and significant contribution. The comments and suggestions have been included in the revised manuscript. Please see the revised manuscript.

Reviewer 2 Report

This paper is focusing on determination of kinetic energy released during the quasi-static fracture of granite rock. The elastic and total energies stored in the specimens were obtained through the quasi-static compression curve; and the released kinetic energy was obtained through high-speed camera photography. In overall, the paper is interesting. However, the authors must correct the manuscript according to some major and minor comments appearing in the attached file.

Author Response

(The authors gave the same response as above.)

Reviewer 3 Report

Dear Authors,

The reviewed paper address a very important issue of materials fracture. Dynamic fragmentation has been studied extensively using different methodologies as you mentioned in the paper. The paper describes very well the high-velocity recording methodology to measure the fragment's kinetic energy losses in the breakage process, however, a comparison of the results with another measurement method as the Split Hopkinson Pressure Bar is missing.

Additional comments are included in the PDF file.

Regards

Author Response

(The authors gave the same response as above.)

Round 2

Reviewer 2 Report

The manuscript can be published